# Raising Voice at School: Preliminary Effectiveness and Community Experience of Culture and Practice at an Australian Trauma-Responsive Specialist School

**Julie C. Avery [1,*]** , **Emma Galvin [1]**, **Joanne Deppeler [2]**, **Helen Skouteris [1,3]** , **Justin Roberts [4]** and **Heather Morris [1]**

1   Health and Social Care Unit, School of Public Health and Preventive Medicine, Monash University, Melbourne, VIC 3004, Australia; emma.galvin@monash.edu (E.G.); helen.skouteris@monash.edu (H.S.); heather.morris@monash.edu (H.M.)
2   School of Education, Monash University, Melbourne, VIC 3800, Australia; joanne.deppeler@monash.edu
3   School of Business, Warwick University, Coventry CV4 7AL, UK
4   MacKillop Education, MacKillop Family Services, South Melbourne, VIC 3205, Australia
*   Correspondence: julie.avery@monash.edu; Tel.: +64-0210554628

**Abstract:** The aim of this study is three-fold: (1) to explore multiple community member views of core elements of trauma-responsive practice at a specialist school; (2) to explore impact on student wellbeing and learning outcomes, and educator experiences of their workplace; and (3) to explore insights into implementation challenges and enablers. This study uniquely incorporates four participant cohorts: parents or caregivers, educators, and community agencies involved with school students and their families. It utilises a mixed-methods approach with an emphasis on the voice of participants and their lived experience of a trauma-responsive specialist school. The data identify a trauma-responsive school culture, high staff satisfaction, improved student wellbeing and attendance, and progress towards learning goals. Reflective analytic themes centre on a collective experience of the school as a connected community, emphasising relationships, safety, collaboration, mutuality, voice, and empowerment. Findings show that the practices most valued across the cohorts centre on the collective experience of the school as a connected community, emphasising relationships, safety, deep listening, collaboration, mutuality, voice, and empowerment. Trauma-informed principles frame the discussion and implications for equity-focused trauma-responsive practice and policy development. Implications for practice and policy development are discussed.

**Keywords:** educational equity; relational pedagogy; trauma-responsive schools; stakeholder collaboration; voice; empowerment; wellbeing

## 1. Introduction

Almost three out of four children in Australia (similarly in the US and the UK) have been exposed to potentially traumatic events [1]. The term "potentially traumatic" acknowledges that adversity is experienced within the context of the individual, such that the presence of adversity does not necessarily predict whether or not an individual experiences a negative impact or the degree of impact. Individual context includes supports, relationships, personal strategies, historicity, and strengths that interact uniquely for individuals. Contextual issues, such as poverty or systemic injustice, add to the allostatic load (accumulation of stressors) carried by students and communities, which impedes wellbeing. In Victoria, 12.1 per cent of children were the subject of one or more substantiations of family harm, such as exposure to violence, neglect, or abuse [2,3], and approximately 1 in 6 children live in poverty and are socially and educationally disadvantaged at school [3,4]. Children and young people around the world are reporting ever-increasing mental health concerns and effects of adversity, including increased exposure to domestic violence; community violence; and social inequities including poverty,

food insecurity, and homelessness [1,4–7]. Systemic trauma arising from bias, institutional child-abuse [8], colonisation, and knowledge sovereignty [3] perpetuate and increase the trauma load for students already disenfranchised by "othering", related to characteristics of race, (dis)ability, sexuality, historicity [5–7]. The Australian students at the highest risk of trauma exposure are Indigenous, users of child protection or youth justice services, or are refugees/asylum seekers [1,2,4]. Distressing reports from the Australian Royal Commission enquiry into sexual abuse of children while in institutional care further demonstrate the urgency to protect children and support healing [9]. Trauma elevates neurobiochemical stress responses throughout the body. These can remain active and toxic throughout adulthood, adversely impacting health, mental health, relationships, employment, and contribute to poverty and homelessness [5–7,9,10]. The younger the infant or child, the higher the risk and degree of developmental harm from trauma, such as effects on the brain architecture, health, and social responsivity [8,10–13].

Educators are expressing ever-increasing interest in trauma-responsive (TR) school-wide approaches, as they seek to improve learning and wellbeing outcomes for students impacted by adversity and the adults that support them [5,7,14,15]. Trauma, as defined by the Substance Abuse and Mental Health Services Administration (SAMHSA), results from an event, a series of events, or a set of circumstances that is experienced by an individual as physically or emotionally harmful or life-threatening and that has lasting adverse effects on the individual's functioning and mental, physical, social, emotional, or spiritual wellbeing ([16], p. 7). Trauma-informed initiatives began in adult mental health and health care [16] and have since been adapted and adopted across health, education, and social care services over the past two decades [6,14]. Recent TR models in education centre on relationality; addressing inequities; and responding to systemic, racial, cultural, and historic trauma [3,5,7].

School settings can provide numerous protective factors for students impacted by trauma, such as academic achievement, supportive student–teacher relationships, and social connectedness [14,15]. Safety, relationality, and sense of belonging are fundamental to recovery and healing from trauma and are supported by moment-to-moment interactions, focusing on empathy, building trust, belonging, and social–emotional capacity [7,10,14,15]. The provision of optimal wellbeing and learning conditions to meet student needs and promote social equity is a fundamental aim of the recently revised Australian Curriculum [17], aligning with priorities highlighted in United Nations [16] sustainable Goal 3, to ensure health and promote wellbeing, and Goal 10, to reduce inequalities. Nevertheless, Berger et al. [18] found that Australian teachers lack confidence and feel under-prepared to work with students impacted by trauma.

Environments aiming to support wellbeing, healing, and post-traumatic growth and prevent further harm must (a) realize the widespread impact of trauma and potential paths for healing; (b) recognize the signs and symptoms of trauma; (c) resist (re)traumatization; and respond by integrating trauma knowledge into practice, policy, and procedures [19]. The SAMHSA offers six key principles of trauma-informed practice: (i) safety; (ii) trustworthiness and transparency; (iii) trauma-impacted peer support; (iv) collaboration and mutuality; (v) empowerment; and (vi) responsiveness to cultural, historical, and gender issues. Translating TR principles into the fabric of a school is critical to establishing a TR climate [6,10,15,20]. Measures of school climate, such as Attitudes Related to Trauma-Informed Care (ARTIC), [21], can help to prioritize TR implementation, evaluation, and the planning of TR educator professional development. Attitudes and beliefs related to student behaviour, discipline, and student potential have been found to be important drivers of TR change [10,20,22,23], as has the influence of leadership on school change initiatives, school climate, educator motivation, and practice [23–26].

Whilst understanding educator perspectives regarding TR practice is essential, it has dominated TR school research, leaving a considerable gap in appreciating what TR practice may include from the perspectives of multiple members of the school community. Inclusive consultation is central to improving knowledge and the understanding of what

TR environments look like, feel like, and mean from the perspectives of those holding less power to influence school practices: students, guardians, and external agencies. A rapidly growing body of work positions equity and empowerment as fundamental for TR environments [3,5,7,27], particularly as schools have been sites of trauma, perpetuating inequities, disenfranchisement, and harm [6,7,9]. Hearing the voice of the school community and understanding the lived experience of a school from diverse perspectives could anchor TR development to what is meaningful to those whom schools serve. By proactively valuing the voice of children, families, and communities, research could improve the understanding of experiences that shape students' lives and approaches at school that benefit them; our study aims to contribute to this understanding by evaluating outcomes and stakeholder experiences with an emergent TR model.

### 1.1. The Reframing Learning and Teaching Environments (ReLATE) Model

ReLATE is an Australian inquiry-based school model aiming to create optimal environments for learning and teaching and enable healing from adversity. ReLATE evolved within a specialist school governed by a large community services organisation that implements the Sanctuary Model [10,28] of trauma-informed practice across their various social care services. ReLATE is informed by trauma theory, principles, and practice frameworks [6,25–29] and utilizes components common to TR practices [29], including trauma training, a focus on safety, relational connection, and adapted policies and procedures. ReLATE practices are strongly informed by the Sanctuary Model; Therapeutic Crisis Intervention in Schools ([30], TCI-S); Positive Education [31,32]; Visual Learning [33]; relational pedagogy, social, and systems theory [6,10,34–36]. The TCI-S Life-Space interviews and debrief process are weekly components of school practice, as ReLATE continues to iteratively adapt and contextualize practice to various school contexts [37–39]. This study was undertaken during the sixth year of model development and explores the reach and impact of the emergent ReLATE model during the period January to December 2019 to inform its ongoing development. Figure 1 illustrates the core constructs of the ReLATE model.

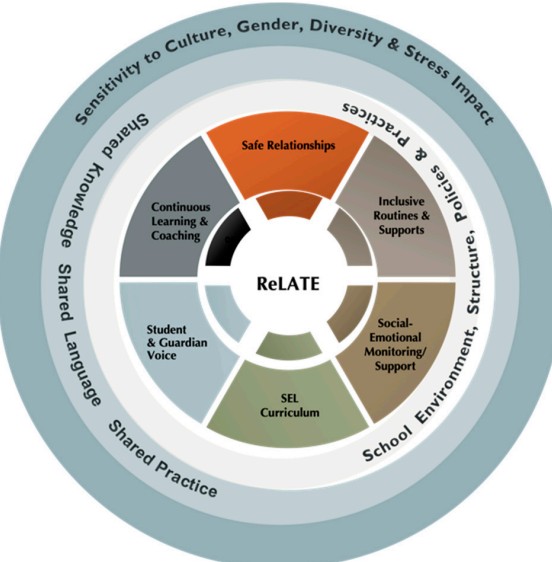

**Figure 1.** The ReLATE model of trauma-responsive practice.

The four foundational elements of the model are outlined in Figure 2.

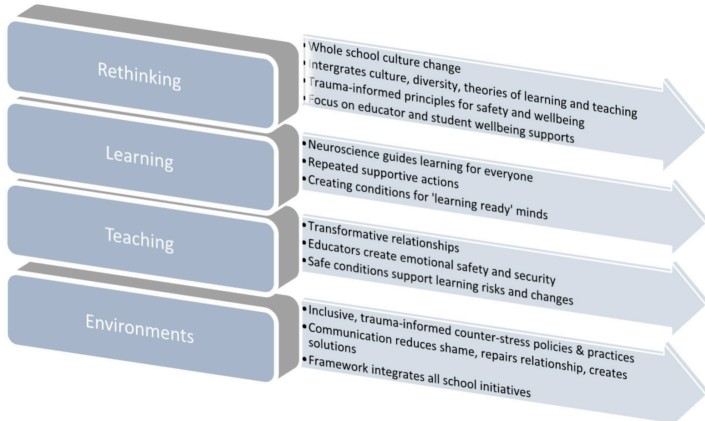

**Figure 2.** ReLATE's 4 foundational elements (adapted with permission from The MacKillop Institute [40]).

### 1.2. Purpose of the Study

This study responded to the urgent need to address the potentially life-long impact of childhood adversity, and the global increase in the mental health and wellbeing needs of students [1,2,4,20] and their educators [14,15,18] through the implementation of relational, neuroscience-informed, and trauma-responsive practices. Applying a school-wide multi-stakeholder approach to the inquiry, the aim of this study was two-fold: firstly, to evaluate the preliminary effectiveness of the ReLATE model in improving student outcomes; secondly, to understand the aspects of ReLATE that were valued from the perspectives of key stakeholders with lived experience of the school, i.e., students, guardians, educators, and community agencies. The specific research questions were the following:

1.  What is the evidence of improved student attendance, wellbeing outcomes, and academic progress?
2.  How is the school experienced by the school community, i.e., students, guardians, educators, and agencies?

### 2. Materials and Methods

#### 2.1. Study Design, Ethics, and Informed Consent

To address the research questions, this mixed-methods participatory action study employed surveys, academic and wellbeing data, school documentation, interviews, and focus groups. As TR approaches require system-wide implementation [6,19,40,41], data were collected from individual to organizational levels. The Consolidated Criteria for Reporting Qualitative Research (COREQ) checklist [42] guided this study. Ethics approval for the study was obtained from the Monash University Human Research Ethics Committee (MUHREC)—7826—and from the community organisation's Ethics Review Committee. Participants were provided with a cohort-specific explanatory statement and consent form in advance of data gathering. This was achieved either via email or hard copy. The researchers ensured that there was written informed consent from each participant, including consent to publish findings. Following receipt of guardian consent to their child's participation, students were asked to assent before beginning an interview or focus group. The voluntary nature of participation was emphasised. Data gathering was conducted over the period of January 2019 to January 2020.

#### 2.2. Setting

This study took place at a small, urban, non-government specialist school in Victoria, Australia, delivering the State Curriculum (Foundation [F]–10) and the Victorian Certificate of Applied Learning (VCAL) to years 11–12. The students lived in a number of urban communities with high levels of socio-educational disadvantage which included those outside the immediate zone of the school. The parents/guardians of students at the school

were overrepresented in low-paid and lower-skilled industries and occupations, a lower level of year 12 completion and university qualifications, and a high rate of part time work. The school caters to students requiring extensive classroom adjustments equivalent to Tier 3 or 4 supports in a multi-tiered system of support (MTSS), utilising high-impact, evidence-informed pedagogical practices to provide tailored, appropriate levels of support and adjustments for individual students.

The student cohort included learners with a disability, neurodiversity, and/or mental health diagnosis, many of whom had had significant periods (>1 year) out of school. Student enrolments at the time of this study totalled 64, with 21.88% identifying as Aboriginal or Torres Straight Islanders. The school aims to develop social–emotional capacities and improve learning outcomes and transitions students back to mainstream (general) schools or on to tertiary education or employment training. All staff have been trained in TR practices, receive ongoing professional learning development (PLD), and engage in peer support and supervision. The school utilizes a coaching–mentoring approach to learning and teaching and describes its culture as one of "unconditional positive regard" with an expectation that "every student will do well if they can". An already low student-to-educator ratio at the school was further reduced at the start of 2019 to 1:4 (1 teacher and 1 education support worker) in all classrooms.

### 2.3. Researchers, Participants, and Recruitment

2.3.1. Researchers

The research team consisted of 4 females: a developmental psychologist and senior researcher ($n = 1$); educational psychologists ($n = 2$), and two health and social care researchers. Author 1 (New Zealander, educational and developmental psychologist) and Author 2 (Australian, health and social care researcher), experienced in qualitative research, were based at the school for a week to conduct interviews and facilitate focus groups during and after school hours. Author 1 was primarily responsible for data collection, analysis, and reporting, which she approached through a neuroscience-of-learning-and-trauma lens, informed by socioecological, systems, and social equity theories, combined with extensive experience working with marginalized children, youth, and families with diverse and complex needs within a range of school, child protection, and social care systems.

2.3.2. Participants and Recruitment

There were four participant groups: internal stakeholders (educators, students, guardians) and external stakeholders (community and government agencies). Educators included administration and leadership/senior management, teaching, and non-teaching staff (wellbeing team, education support staff, and IT personnel).

1.  Educators. A general outline of the study was emailed to the staff by the principal, with a follow-up invitation to participate in the study offered by the researchers at an all-of-staff meeting. The voluntary nature of the study was highlighted, and the explanatory statement and consent forms were distributed with an expression-of-interest form, interview, and focus group schedules.
2.  Guardians. Guardians were first notified, via the school newsletter, that researchers would be in the school and that there would be an opportunity to participate in the study to inform ongoing school practice development. The study was further introduced to guardians by staff from the school wellbeing team (reading from a script provided by the researchers) at the conclusion of each student's regular learning support group meeting held at the beginning of the second school term; an expression-of-interest form was available so that contact details could be provided to the researchers.
3.  Students. Students were all enrolled at the school in grades F-12 (ages 6–18 years). Students were introduced to the study, alongside their guardians as outlined above. Students whose parents had consented to them participating were required to give their individual assent to participate. There were no exclusionary criteria.

4. Agencies. A list of agencies, such as disability services and child and adolescent mental health services, that are regularly involved with students, the referring mainstream (public) schools, and the Catholic Education Office school governing body was provided by the wellbeing team co-ordinator. Initial contact was made via email, outlining the purpose of the study, the research team, and contact details for queries and options to register interest.

### 2.4. Materials and Procedures

A range of tools were used to address the research questions as illustrated in Table 1 and listed below.

**Table 1.** Research questions, data gathering tools, and participant group crosswalk.

| | Measures Used to Answer Research Questions | | | | | | | |
|---|---|---|---|---|---|---|---|---|
| | Interviews | Focus Groups | ARTIC-ED45 | [1] SDQ | [2] EES | [3] DET | [4] CEMSIS | Wisconsin DI [5] Review Tool |
| **Research Question** | **Participants: Students (S), Guardians (G), Educators (E), Agencies (A); School Documentation (D)** | | | | | | | |
| Q. 1 What is the evidence of improved student wellbeing outcomes and academic progress? | S; G; E; A | S; G; E | | G | | S; G; E | S; G; E | D |
| Q. 2 How is the school experienced by stakeholders, i.e., students, guardians, educators, and agencies? | S; G; E; A | S; G; E | E | | E | S; G; E | S; G; E | D |

[1] Strengths and Difficulties Questionnaire, [2] Employee Engagement Survey, [3] Department of Education and Training Review, [4] Catholic Education Melbourne School Improvement Survey, [5] Department of Instruction Wisconsin: Policies/Procedures Review Tool.

Quantitative Data Gathering

1. Attitudes Towards Trauma-Informed Care (ARTIC) Scale

Building trauma-responsive organisational climates is central to sustaining trauma-responsive practices and healing environments [6,10,22,26]. Although it is important to measure school climate, there are few validated tools for schools to utilise. The ARTIC is one psychometrically valid measure of professional and para-professional attitudes toward trauma-informed care (TIC) with excellent internal reliability ($\alpha = 0.93$) [21]. The ARTIC-45 Education (ED) scale is a self-report measure used to evaluate a range of constructs relevant to TIC, is designed to be used in schools already trained in and implementing TR practices and can be used for quality improvement. The ARTIC produces a total score ($\alpha = 0.92$) along with scores for seven subscales (reliability range $\alpha = 0.68$–$0.75$), as described in the categories provided in Table 2. Higher scores indicate stronger positive views towards trauma-informed care principles and practice. Percentile ranks indicate how the school compares to scores in the validity studies for the scale.

**Table 2.** The ARTIC-45 scale subsections.

| | ARTIC-45 ED Subscale | Measure of School Personnel Attitudes |
|---|---|---|
| 1. | Underlying causes of problem behaviour and symptoms | Are student behaviours and symptoms viewed as adaptive and malleable or intentional and fixed? |
| 2. | Staff responses to problem behaviour and symptoms | Should responses to problem behaviour focus on relationship, flexibility, kindness, and safety as the agents of change or focus on accountability and consequences? |
| 3. | Empathy and control | Should staff be empathy-focused versus control-focused? |
| 4. | Self-efficacy at work | Do staff feel able and confident to meet the demands of working with traumatized students, or do they feel unable to meet the demands? |
| 5. | Reactions to the work | Do staff appreciate the effects of secondary trauma and cope by seeking support, or do they minimize the effects of secondary trauma and cope by ignoring or hiding the impact? |
| 6. | Personal support for trauma-informed care | Do staff feel supportive of and confident about implementation of TIC versus concerned about implementation of TIC? |
| 7. | System-wide support for trauma-informed care | Does the wider system (e.g., administration, supervisors, colleagues) support TIC, or does it not support TIC? |
| | Total mean score | Reflecting answers to all items of the ARTIC scale, this is a global score of attitudes toward TIC. |

2.    Strengths and Difficulties Questionnaire (SDQ—Parent Form)

The SDQ [43] is considered a reliable and validated brief (25-item) measure of wellbeing, adjustment, and/or psychopathology of children and adolescents. The reliability scores for the SDQ are based on Australian data and psychometric properties of the Strengths and Difficulties Questionnaire [44,45], which indicate moderate-to-strong internal reliability and stability across all SDQ subscales. Alpha coefficients ($\alpha$) for each of the five SDQ subscales and the total difficulties and impact scales range from $\alpha = 0.59$ (peer problems) to $\alpha = 0.80$ (hyperactivity). Adequate validity was evidenced in the relationships among these scales. SDQ scores are predictively valid, evidencing the feasibility of the SDQ as a screening instrument with concurrent validity against diagnostic interviews, ranging from 0.12 to 0.57. The alpha coefficients for the SDQ subscales range from 0.65 to 0.91 for each of the subscales. McDonald's omega, or Jöreskog's rho, ranges from 0.67 to 0.90 for the parent version [46,47].

The Parent Form of the SDQ was used to examine the school impact on student wellbeing from the perspective of parents/guardians, unlike other TR studies which focus on teacher views [8,29], and to better understand whether any impact on student wellbeing was evident outside of the school environment (i.e., at home or in the community). SDQ data were gathered for a subgroup of students ($n = 18$) in the current study at three timepoints: at baseline, Timepoint 1, December 2018; Timepoint 2 at 6 months; and Timepoint 3 at 12 months, December 2019. Data gathering was led by the school psychologist and overseen by Researcher 1. The procedure, analysis, and findings were published [48]. The guardians consented to the researchers' use and publication of this information.

3.    Documentation: Reviews of Policies, Procedures, and Practice Manuals

Following completion of interviews and focus groups, the school administration provided access to a range of online documentation ($n = 17$), including school mission,

staff onboarding, discipline, safety protocols, strategic plan, mandatory staff training, practice manuals, de-identified attendance rates, incidents, and Individual Education Plans, along with survey and evaluation reports, as noted in Table 1. School operating systems, such as policies, procedures, and practice manuals, that align with TR principles are critical to implementing and sustaining TR practice [23,40,49–51]. Documents were uploaded into NVivo and deductively coded against the Wisconsin Review Tool for School Policies, Protocols, Procedures & Documents: Examination Using a Trauma-Sensitive School Lens [52], which is framed by the five practice domains for trauma-informed care as described by Harris and Fallot [43], that is, safety, trustworthiness, collaboration, choice, and empowerment, and considers the degree to which TR principles and practices are reflected in the language and recommendations of school documentation and systems.

4.     Employee Engagement Survey (EES)

EES data were made available to us by the community organisation. The survey was conducted by Best Practice Australia Ltd. [53] at the commencement of school in 2014; in year 2, 2015; and in March 2019, the sixth year of designing and implementing TR practices. The engagement survey uses a benchmarking process against norms in the school sector to report organisational culture on a continuum from Blame, Reactivity, Consolidation, Ambition to Success. The survey was sent to all staff via Survey Monkey, with staff emails being provided by the community organisation. For additional information on the survey itself, see BPA Analytics [53].

5.     Department of Education and Training (DET) Review 2019

The Department reviews school programs on a 3-year cycle using independent evaluators to determine performance against the elements that identify highly performing school environments. The review includes a combination of student, guardian, and staff surveys; interviews; classroom observation; and review of school documentation and student outcome data [54].

6.     Catholic Education Melbourne School Improvement Survey (CEMSIS)

The CEMSIS [55] is a 4-yearly school improvement review by Catholic Education Melbourne using a set of online surveys built specifically by BPA Analytics [53] for Catholic schools to ensure that the values that inform a school's vision are translated into best practice in child safety, learning and teaching, student wellbeing, community, leadership, and management. The review includes surveys for educators, students, and guardians, and school documentation and outcome data were conducted at the commencement of school in 2014; in year 2, 2015; and in March 2019, the sixth year of designing and implementing trauma-responsive practices.

*2.5. Qualitative Data Gathering*

Interviews and focus groups were conducted at the school during school hours at the beginning of Term 2 (May 2019); in addition, there were two evening focus groups for guardians. Agency interviews (20 min average) were conducted over the phone in Term 4 (October/November 2019). Interviews ranged from 30 to 40 min, and focus groups, from 60 to 75 min. Student interviews averaged 15 min, although less for younger students, and focus groups averaged 25 min. During all focus groups, Author 1 facilitated each session, with Author 2 taking notes on non-verbal aspects (e.g., nodding agreement), key points, repeated ideas, words, phrases, and progress of the discussion. Researcher debriefs were conducted immediately following each focus group, with reflective journaling and field notes being documented. Interviews and focus groups followed the same semi-structured process and line of questioning into perceptions and elements of the school culture and practice that were valued. Examples of the questions used in interview and focus groups are provided in Supplementary Table S1. Follow-up questioning was utilised to encourage participants to expand on their responses as necessary and reflect on their experience of the

school and the approaches used. All focus group participants were encouraged to share their views and contribute to the discussion.

Analysis

The ARTIC–45ED survey data were imported into a pre-formatted Excel spreadsheet (ARTIC-QSR) designed to compute the mean subscale scores. Data were then collated using the median of the school-wide scores for each of the subscales. The scores are divided into the following three benchmark ranges based on percentile rank: Thrive range—75th to 100th percentiles; Grow range—25th to 75th percentiles; Learn range—0 to 25th percentiles. To evaluate any emergent effects of ReLATE on student SDQ scores, a series of repeated-measure analyses of variance (ANOVAs) were carried out. The ANOVAs explored whether significant adjustments occurred over a 12-month period. This analysis was followed up with the Reliable Change Indicator analysis to determine effect size and to distinguish whether a pre-to-post-treatment score difference was meaningful or was a random error [56,57]. SDQ data gathering and analysis were facilitated by the school psychologist and overseen by the first author.

*2.6. Qualitative Data*

Interviews and focus groups were audio-recorded, professionally transcribed, reviewed for accuracy while listening to recordings, and imported into NVivo (version 12.0) [58] for coding and analysis. This study used a reflexive thematic analysis approach that was suited to the emergent nature of research in school-wide TR practice and the intention to understand participant experience. Transcripts were imported into NVivo 12.0 prior to coding. In the early phase of analysis, the focus was on an inductive process with the semantic coding and noting of candidate themes for further exploration. The first author carefully read the transcripts and simultaneously listened to the recordings, re-read, and reflexively considered the transcripts, a process that included journaling thoughts and comments and creating mind maps. Coding began with capturing the overt meanings of the participants, exploring the data for codes and candidate themes and sub-themes. Author 1 coded all manuscripts, and Author 2 independently coded 20% of the transcripts to explore and further discuss alternative meanings with Author 1. This approach is in accordance with the most recent recommendations by Braun and Clark [59], who argue that a quantitative construct of validation and reliability does not meaningfully apply to big Q (qualitative) methodologies.

The data analysis was a cyclic process of reading, reflecting, stepping back from, re-reading, discussing data with the research team. This process was determined to be optimal when exploring underpinning or alternative meanings, contradictions, and comparisons across stakeholder groups. Codes and themes were iteratively revised and adjusted; coding became increasingly latent and deductive, considering that conceptual meanings as strong links to trauma-informed principles, theory, and research were noticed in the data [59]. Familiarity with the topic, interdisciplinary training, and experience led Author 1 to take an inquiry-based and interpretive approach to data collection and all phases of analysis across 16 months.

Early preliminary findings from the first 6 months of analysis were presented at a wellbeing conference; further refined and deeper-level results were presented during a webinar and an educator conference prior to writing up the report. These presentations provided additional insights into the data and what resonated (or not) with educators, researchers, and peers. We cite participant quotes liberally to ground the qualitative data in the participant's lived experience and personal meaning making.

## 3. Results
*3.1. Participant Demographics and Characteristics*

Table 3 presents participant demographics. A total of 47 interviews and 9 focus groups were conducted, for a combined total *N* = 91. Apart from two staff, all educators (E) had

participated in 2-day Sanctuary Model training in addition to Cornell University 4-day Therapeutic Crisis Intervention (TCI) training. TCI is a staff training programme [31]. All staff received ongoing trauma-responsive development. Guardians (G) included parents and foster parents, grandparents, or caregivers of a student(s) enrolled at the school.

**Table 3.** Participant demographics.

| **ReLATE Study Participant Demographics** | | | | | | | | | |
|---|---|---|---|---|---|---|---|---|---|
| **Variables** | **Categories** | **Educator N = 26** | **Educator %** | **Guardian N = 26** | **Guardian %** | **Agency N = 6** | **Agency %** | **Student N = 22** | **Student %** |
| **Gender** | Female | | 62.5 | 24 | 82.6 | 4 | 66.67 | 5 | 23 |
| | Male | | 37.5 | 2 | 17.4 | 1 | 16.67 | 17 | 77 |
| | LGBTIQ+ | 0 | 0 | 0 | 0 | 1 | 16.67 | 0 | 0 |
| | Not specified | 2 | 0 | 0 | 0 | 0 | 0 | 0 | 0 |
| **Ethnicity** | Australian Caucasian | | | 22 | 84.62 | 4 | 66.67 | 16 | 72.73 |
| | Australian Aboriginal or TSI | 0 | 0 | 2 | 17.4 | 1 | 16.67 | 3 | 13.64 |
| | European (German, Serbian, British) | 3 | 11.11 | 1 | | 1 | 16.67 | 1 | 4.55 |
| | New Zealand Caucasian | 1 | 3.7 | 0 | 0 | 0 | 0 | 0 | 0 |
| | Not specified | 2 | 7.41 | 2 | 17.4 | 0 | 0 | 2 | 9.09 |
| **Age (years)** | ≤29 | 5 | 18.52 | 1 | 3.87 | 0 | 0 | Age | |
| | 30–39 | 9 | 33.33 | 6 | 23.08 | 2 | 33.33 | 6–9 | 27.27 |
| | 40–49 | 9 | 33.33 | 15 | 57.69 | 3 | 50 | 10–14 | 31.82 |
| | ≥50 | 4 | 14.81 | 4 | 15.38 | 1 | 16.67 | 14–18 | 45.45 |
| **Teacher qualifications** **All teachers certified with Bachelor of Education minimum qualification** | | | | | | | | | |
| Teaching experience (years) | ≤4 | 2 | 7.41 | | | | | | |
| | 5 to 9 | 9 | 33.33 | | | | | | |
| | 10 to 14 | 9 | 33.33 | | | | | | |
| | 15+ | 5 | 18.52 | | | | | | |
| Level of trauma training 1–7 (1 = brief & little follow-up, 7 = very extensive & ongoing support) | <3 | 0 | | | | | | | |
| | 3–5 | 37.04 | | | | | | | |
| | 5–7 | 62.96 | | | | | | | |
| **Trauma training/support satisfaction** | | | | | | | | | |
| 1 (low)–7 (high) | 1 to 3 | 0 | 0 | | | | | | |
| | 4 to 5 | 8 | 29.63 | | | | | | |
| | 6 to 7 | 19 | 70.37 | | | | | | |

Some households had more than one child at the school; the participation rate was 85.19%, representing 38.33% of guardians of students enrolled at the school. One-third of the student (S) population participated in the study, with an almost equal spread of primary (*n* = 11) and secondary students (*n* = 10). All students who indicated intent to participate engaged in an interview/focus group. Of the seven agencies (A) invited to participate, six agreed and were interviewed, with a participation rate of 85.71%.

*3.2. Quantitative and Qualitative Findings*

The two-fold aim of this study was to evaluate the emergent effectiveness of the Re-LATE model to improve student outcomes and understand how the school was experienced by key stakeholders. Findings are presented below.

Research Question 1. What is the evidence of improved student attendance, wellbeing outcomes, and academic progress?

Student Outcomes

The outcomes of the SDQ across a 12-month period indicated a decrease in behavioural, mental health, and wellbeing concerns from the perspective of their parent/guardian(s). Student attendance and learning progress outcomes are reported in Table 4. A large effect size in mean total difficulties, a decrease of 29%, indicated a clinically significant improvement in wellbeing and mental health concerns, with the greatest improvement being seen for students in their first year at the school. Of the students who were new to the school that year, 37.5% had a diagnosis of ADHD. Hyperactivity in this group decreased at a consistent rate throughout the 12-month period, with the mean hyperactivity score reducing from a classification of "clinical" to "slightly raised". Over 12 months, a significant reduction in conduct problems (F = 6.76, *p* = 0.04), hyperactivity (F = 11.39, *p* = 0.01), and total difficulties (F = 12.07, *p* < 0.01) was reported, along with a close to significant reduction in mean scores for emotional symptoms (*p* = 0.06), reducing from above a clinical range to below the clinical range post-intervention [49]. As the SDQ data have already been published, we refer the reader to Diggins [49] for a more detailed analysis and discussion. The CEMSIS and EES do not have analytics in the public domain at this time.

**Table 4.** Study measures and outcomes.

| Measure | Item | Percentage |
|---|---|---|
| School statistics | Staff retention<br>Staff attendance | 86.5<br>94.4 |
| Employee Engagement Survey | Level of engagement<br>Mid-2014 (year 1) Culture of Consolidation<br>Mid-2015 (year 2) Culture of Success<br>Mid-2019 (year 6) Culture of Success | (% of engaged staff)<br>42<br>62<br>88.7 |
| CEMSIS [1] 2019 (April–May)<br>Educator views | 1. Staff have a positive perception of the relationship between staff members and the leadership team.<br><br>2. Staff believe that school leaders set conditions for improving learning and teaching.<br><br>3. Staff feel safe to take risks and make mistakes at the school. | 85<br><br>75<br><br>80 |
| Student views | 1. Students report positive student–teacher relationships.<br>2. Students feel they truly belong/are valued and are cared about in the school community.<br>3. Students feel there are rigorous expectations of them. | 71 (P [2]); 51 (S [3])<br>72 (P and S)<br>62 (S) |
| Guardian views<br>Primary age child (P)<br>Secondary age child (S) | 1. School positively overcomes barriers to student engagement.<br>2. School has a positive social and learning climate.<br>3. School provides timely, frequent, and effective communication.<br>4. School matches their child's developmental needs. | 85 (P); 82; 82 (S)<br>81 (P and S)<br>80 (P); 84 (S)<br>72 (P and S) |
| DET [4]<br>(July 2019–Jan 2020) | NB: *Regular* attendance at school = attending 90% of the school year<br>Student attendance at previous school (AVE across student population)<br>Student attendance after 12 months at the school (AVE)<br>Students with Individual Education Plans (IEPs) with 4 key goals<br>Achievement or progress towards planned outcomes/goals<br>1379 IEPs created; 965 goals successfully achieved<br>Academic goals achieved<br>Attendance goals achieved<br>Transition goals (to mainstream school, tertiary education, or employment training)<br>Engagement goals | %<br>56<br>83<br>100<br>70<br>70<br>65<br>68<br>75<br>72 |

[1] Catholic Education Melbourne School Improvement Survey ([56], pp. 23–25), [2] primary students, [3] secondary students, [4] Department of Education and Training ([55], p. 12).

Educator Outcomes

The ARTIC survey was completed by *n* = 26 educators, a participation rate of 92.8%. Results from the survey (Figure 3) indicate that the school established a strong trauma-responsive culture, showing substantial strengths within the Thrive range (75th–100th percentile), for understanding the underlying causes of behaviour; focusing on the importance of relationships, flexibility, kindness, and safety as the agents of change; being empathy-focused; personally supporting the implementation of trauma-informed practices; confidence in implementing and feeling that the school actively supports the use of trauma-informed practices. Self-efficacy at work and reactions to the work ranked in the Grow range (25–75%), with 5.17, indicating that staff had a level of confidence (just below the Thrive range) in meeting the demands of working with trauma-impacted students and an appreciation of the effects of secondary trauma and could seek support versus minimizing the effects of secondary trauma/vicarious trauma. The outcomes from the EES, CEMIS, and DET data gathered throughout 2019 (Table 4) are consistent with the ARTIC outcomes found in our study.

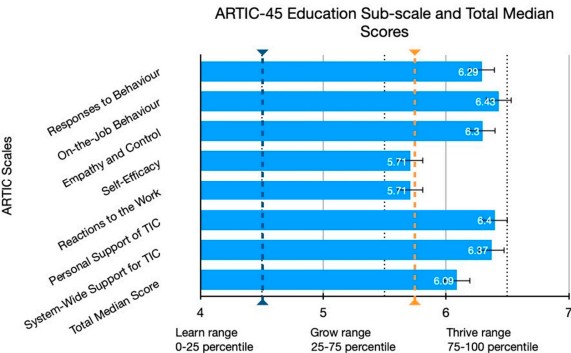

**Figure 3.** Educator ARTIC-45 subscale and total median scores. Error bars indicate the margin of error, a half-width of the 95% confidence interval of the data-point.

School Outcomes

A total of *N* = 528 references to TR practice elements were made across 17 school documents (mean = 35.066; Med = 18), placing the majority of school policies and procedures within the highest quartile ((Q) 1 ≥ 9, Q 2 ≥ 18, Q 3 ≥ 56) with a distribution of 2–11 items coded against the Wisconsin Review Tool [51] per document.

The data reported in school statistics, EES, CEMSIS, and DET are reported in Table 4. Overall, the findings are coherent, showing a continuing upward trend of educator engagement, a high teacher retention rate, alignment of school documentation and TR principles, and positive stakeholder opinion.

Key data on student satisfaction from the CEMSIS indicated greater variability and somewhat lower levels of school satisfaction reported by students participating in our study.

In 2019, nine students transitioned back to their mainstream school; five students entered further education or training; six students entered alternative education; and two students entered employment. Guardians reported high levels of satisfaction regarding the impact of the school, staff practices, and improvements in their child's behaviour and emotion management in both the school and the home contexts. The year 12 program developed at the school was recognised by the Victoria Curriculum and Assessment Authority (VCAA) in 2018 and 2019 for high-quality practice and innovation [55], with one year—12 graduate receiving the VCAA Achievement Award for Personal Development Skills.

Research Question 2. How do key stakeholders experience the school?

Two supraordinate themes describe the collective stakeholder experience: (1) Care about us; (2) Power together. Five sub-themes further express stakeholder perspectives: (a)

Collective care, collective understanding; (b) Feelings matter; (c) Trust; (d) Tell me what I do well, and plan with me to do better; (e) "I am hopeful for the future".

Given the quantity of qualitative data, only key representative quotes from each stakeholder cohort are used in each section, with additional quotes, educator practices, and elements valued by stakeholders in this study (see Supplementary Table S2).

Supraordinate Theme 1. Care about us (empathy, attuned connection, and community)

Students across primary and secondary levels felt that all teachers at the school where supportive and took time to get to know them; they consistently referred to feeling understood and known as an individual, as stated by this primary student: *"teachers understand us more here. . . and I feel happier coming here"* (S7). Students spoke of being able to get to know educators and connect with them: *"At lunch they just talk to us about stuff"* (S3), and *"people can communicate easier because there's more connections here"* (S1). Students, without exception, identified at least one adult in the school whom they could go to and receive support from in times of stress. The theme of connection was reiterated by guardians, as expressed by G1: *"They know the child, they talk to the child. They talk about his interests, his home life. The child feels important"*, and *"staff build a rapport with the students"* (G9). Likewise, agencies stated, *"It feels very community connected"* (A1). *"They display alot of empathy with the students"* (A5). Educators emphasised the need to listen to students telling their story: *"there's a deep commitment for every staff member to get to know their student as deeply as they possibly can"* (E1); they also emphasised the need to build relationships as central to increasing student emotional and learning capacity. As a leader in the school expressed, *"The support we show for each other is modelled for the multi-relationships we're wanting to make accessible to the students, to be able to build in their own lives"* (E3).

Although students were referred to the school primarily due to significant behavioural dysregulation, an overriding sense of safety was collectively expressed by participants, even when students were struggling, as articulated by this senior student: *"I don't have that many safe places but this is a safe place for me, well, I mean totally safe, that I feel like nothing bad is going to happen"* (S10). The importance of the overall school culture is summarised by this agency: *"The school provides students who have experienced adversity with a different experience of the world, adults and relationships and this can change the trajectory of their life"* (A4). Several guardians reflected that the school had a positive impact on them: *"Coming to this school, for me it changed me as well. . . Coming here changes your frame of mind, you become more flexible. . ."* (G21).

Supraordinate Theme 2. Power together (collaboration, mutuality, authentic voice, and empowerment). Stakeholders described a spirit of inquiry and co-creation of meaning at the school, where educators, students, guardians, and community agencies considered together how best to meet individual student needs. Guardians highlighted their experiences of being valued partners in the process of change along with their child, co-determining what successful outcomes looked like and how to reach them: *"I felt so welcomed, they say 'we're here for you, we'll work as a team, if you've got any tips, if you've got any ideas'. . ."* (G4); *"My child is included in his learning plans. Any problems. . . I know I can speak to staff at this school they work with you to help get through them"* (G8). Guardians also described a collaborative and flexible approach to learning and social–emotional plans: *"so we had another meeting and reset things. We all discuss and put ideas out there and, in the end, [the child] decides what he wants"* (G7).

Agencies emphasised how the school empowered students: *"I feel like they're trying to get the student involved in a real active participation in their schooling"* (A2); they viewed school collaborative practices as follows: *"Professionally it's important we can all be around the work with the students. . . I feel like we are providing really good practice, its supporting each other"* (A2). Educators considered this high level of collaboration an important *"point of difference between a non-trauma-informed and a trauma-informed school"* (E5), something that was also observed by the DET [49]. Educators emphasised that authentically valuing voice was core to TR practice, as expressed by this educator: *"It's about providing a space where there's authentic voice from everybody. . . for us it's actually just about voice all the time, continually. [Students*

*are] negotiated with, they're listened to, they're asked what happened instead of what's wrong…"* (E1). This prioritizing student voice was reiterated in the school audit: *"The school creates on-going opportunities to seek students' views on a range of areas of school-life to develop students' recognition that their voice and opinion are valued"* ([54], p. 13). Experiences of collaboration and mutuality were frequently expressed as an overarching sense of acceptance, respect, and dignity by participants, as expressed by guardian G6: *"How [teachers] are here makes you feel very strong. That it's ok to ask questions. My [child] got really good support with his Aboriginal trainer that comes here too. He thoroughly enjoys that, so he can express himself to him"*.

Sub-Themes:

1.  Collective care and understanding. A collective understanding of school practices and a use of shared language was evident in interviews and focus groups. Students across all grade levels explained their personal regulation strategies and described what worked well for them, also describing strategies used by their peers: *"I have friends here and we look-out for each other, we know each-others safety-plan and can help suggest stuff"* (S2). Guardians spoke of educators at the school as being *"teachers that just get-it, whatever they train them in here, they just get it"* (G1). Several guardians differentiated between knowledge and understanding, stating, *"there's a difference between teachers' being taught something and them understanding it"*, and *"it's not just one teacher in this school… the whole school understands [my child]"* (G5). Participants referred to the attitude of unconditional positive regard and expectations of success educators held and the positive impact that had on students, as summed by this guardian: *"compared to other schools [this school] is like apples and oranges to be honest"* (G3).

    Educators spoke of shared understanding as an interconnection of staff beliefs, values, and practice, stating that *"there is something about perspective, your worldview and your experiences that does make a difference"*, as well as a *"back-space, a value-based place teachers [that stay] come from"* (E3). They also spoke of the experience of being in a *"safe to be vulnerable"* team: *"Here I'm working in a true team, knowing that any one of my colleagues will step-in if they see I need support, or just debrief after work, just connect. We do team-tag here to step away from class when need to"* (E9). Agencies reported that the collaboration around meeting student needs was supportive: *"Professionally it's important we can all be around the work with the students… I feel like we are providing really good practice, its supporting each other"* (A2).

2.  Feelings matter (wellbeing and social–emotional capacity). Educators distinguished their focus on wellbeing as follows: *"It's not that academics' is not a priority, it's that your heart would be to make sure the kid is actually coping first, to step back to 'can we get you in school, engaged and safe?'"* (E5). They described paying attention to student and staff personal and collective wellbeing: *"We're aware of transference of anxiety from kids to teachers and that there can be a contagion effect with incidents"* (E2); *"Sticking to the de-brief process I think it's really important to be able to heal and recover… and… wellbeing meetings… to talk about what teachers are dealing with, what they're finding difficult"* (E2). Students spoke with clarity and satisfaction of their improved emotional capacity and ability to name and recognise feelings and use strategies to regulate them when needed. Students commonly referenced the "zones of regulation" and regulation strategies as reasons why they felt good at school, as stated by this secondary student: *"I've learnt how to deal with [stress], like ways we can prevent it from happening when we're sort of getting in the 'zone'"*(S11); similarly, a primary student stated, *"we learn how to tell the teachers, and actually talk about [strong emotions] instead of keeping it in, and it's really calmer and I feel less stress because I have less presure on me"* (S1). Improvement in emotional regulation at home was commonly reported by guardians: *"Even at home, he's different with his emotions. Like, I'll have to growl… but the behaviour will stop. It won't continue for hours and hours and hours, like it used to. His growth, emotionally, has been really good"* (G6).

3.  Trust (consistency, communication, and transparency). Consistency and trustworthiness were viewed as central features of the school, providing routines, predictability, consistent interactions, and follow-through, as expressed by guardian G3: *"The students learn to trust them, and I think the teachers deliver on what they've said that they're going to deliver on, so that provides the student the ability to trust by seeing the teachers can be believed"*. Educators described the "Big 4" of predictability, consistency, routine, and structure as essential practices for building trust, limiting triggers, and creating environments where social–emotional capacity can be developed within individual student windows of tolerance.

Clear, timely, and positive communication was highly valued, as stated by this senior student: *"people can communicate easy because there's more connections; and communication between students is good"* (S14); similarly, guardian G4 stated, *"Communication is really good; email, a lot of one-on-one, I get little phone updates and everything. There's always someone standing out the front of school first thing and in the afternoon"*; this was reiterated by agency A1: *"teachers are quite open with the student and transparent... as part of the [enrolment] process, I felt that they were genuinely interviewing the student, they were really listening to what he had to say"*. Educators emphasised "communication trust": *"It's these ideas of honesty and transparency... not struggle in silence it's like the opposite of that [here]"* (E5); similarly, *"wherever you are in the hierarchy it doesn't matter... that really fosters community"* (E8).

4.  Tell me what I do well, and plan with me to do better (strengths-based, proactive, and preventative). Participants emphasised the positive tone of the school and the proactive building of capacity across emotional, social, learning, and physical capacity, as expressed by a primary student: *"They're more open and instead of giving mean opinions, they give out good opinions. And they teach us how to control our emotions, how to be more kind and more open"* (S6); guardians stated, *"They all have a lot of nice things to say even though my child is not so nice at times. They tell you the positives, and the negatives are put like 'here's what we need to work with you on'"* (G8), and *"it's alright to get it wrong, then we know what you need help with', is the message here"* (G2). All community agencies emphasised how teachers displayed flexible responses, tailoring curriculum demands to the student's cognitive availability on any given day: *"It is also nice to see teachers meeting students where they are at on that day/in that phase of their life"* (A2).

5.  "I am hopeful for the future" (diversity, inclusivity, equity, and access). A secondary student summed the inclusivity of the school with *"We all have our differences and it's ok"* (S2). A felt sense of being respected was commonly reported, as expressed by guardian G6: *"They include [child's] culture a lot, he feels very happy with that"*. Being able to access the curriculum was also viewed as a distinguishing feature of the school: *"They make little steps, like, break it down. Then I can do it and I get to do stuff I enjoy, and I'm interested in"* (S2). Guardians considered that relationships at school contributed to improved attendance and learning: *"It's made a massive change to my child, they're happier to come to school rather than hating school"* (G6); *"My son, he's grown heaps emotionally, physically. His stability's really, really, good. Even at home, he's different with his emotions"* (G10); and *"I think even with the time taken on behaviour my child has learnt more here than at the past school"* (G8). This theme was reiterated by the participants from community agencies.

Students attributed improved outcomes to a number of interconnected factors: feeling understood; relationship with educators; sense of belonging; teaching practices, i.e., learning tasks that were not overwhelming and positive discipline processes that "made sense" and focused on social–emotional development and relational repair; and a calmer overall school environment, as noted in the following comments: *"It's smaller learning tasks and more 1:1"* (S3); *"my teacher understands me, and works on my learning plan so I can actually do it; that's how they show they care—taking the time to always do that"* (S15); *"Basically it's the smaller classes and kind teachers"* (S1); and *"Teachers are actually sitting down with us when we're in a bad mood, and trying to help us, and explaining it; really explaining everything and all that"* (S5). All stakeholders placed class size amongst the reasons teachers understood students. Students spoke of how the small school population reduced feelings of chaos from *"too*

*much going on"* (S8), and in particular, very small class numbers (<10) allowed them to receive more direct support. The view that the school provides an environment of hope and expectations of success is captured by this secondary student's statement: *"I am hopeful for the future... I think [its] how the teachers try and make you become the best you can be"* (S13).

## 4. Discussion

This mixed-methods study responds to the two research questions exploring student outcomes and how the school was experienced by the community members. Findings indicate that the emergent ReLATE model contributed to change for students, staff, and guardians and was overwhelmingly seen as positive by each cohort. Improved student outcomes were evident in school attendance rates, progress towards learning goals and improved social–emotional wellbeing. Our findings describe a school-wide approach that impacted stakeholder experience and student outcomes in important ways. The ReLATE model integrates trauma and learning theories and translates these into practices, ways of being and interacting throughout the school and its systems. Congruent with TR principles, the school positioned understanding safety (as defined from diverse perspectives) and relationality front and centre [4,6,10,29].

### 4.1. Relationality: Centring Connectivity and Optimal Belonging

Relationality was the predominant environmental setting factor in the school that was seen to positively impact outcomes and participant experience of the school. Experiencing connection, belonging, and trust is fundamental to healthy neurobiological development, buffering against the impact of trauma and facilitating healing from psycho-social harm [60,61]. Relationality, connection, and belonging are protective factors for wellbeing, mental health, school attendance, and engagement [60,61]. As Sapiro and Ward ([62], p. 343) reiterate, connection is an "underappreciated and crucial resource for marginalized youth" that is easily undermined by systemic pressures, including class size, curriculum demands, and "key-deliverables" of academic achievement. Conversely, relational disconnection (which punitive and exclusionary discipline contributes to), when not addressed, impedes empowerment and increases detachment from the inner self, further exacerbating student dysregulation.

Succinctly, educators at this school understood that students learn effectively when there are safe and connected learning environments. Additionally, the emphasis on partnering and mutuality in the school suggests knowledge humility and a shift from "educator as expert" to recognizing all members of the school community as knowledgeable contributors to a TR environment. ReLATE supports a culture of collective responsibility and collective care, where all educators share interest in and responsibility for all students and for each other. Psychological safety and the ability to be vulnerable with each other are understood as fundamental to creating "brave spaces to show-up in and learn new skills" [63,64].

Despite the intensity and challenges presented in a school where all students have complex needs unmet by previous mid-to-long-term intervention efforts in former schools, the extremely positive lived experiences of the school, particularly those of parents, educators, and agencies, need to be examined. Influences that may have contributed to the school being held in such high regard may be in contrast with prior negative experiences at other schools. For guardians and students, there is also a reality of very limited viable schooling options available after the mainstream school capacity to improve student social–emotional wellbeing has been "exhausted". The authors further reflected on the data to ascertain if there had been any misrepresentation or skewing of results. Data triangulation within research (i.e., among multiple sources of data) and alignment with external data sources, e.g., from the Department of Education, suggest that stakeholders authentically valued ReLATE practices and the overall school climate. Improved outcomes for students and elements such as the high levels of staff retention and professional quality of life reported here are not able to be attributed to specific elements of the model. Nevertheless, the outworking of the ReLATE model suggests a way of "being together" in schools that is

underpinned by the principles of TR practice [43]. As Cole [9], a pioneer of TR schools, emphasised, TR schools are substantively about how educators are in the space and their translation of principles into practice. Importantly, this study makes a unique contribution to understanding multiple stakeholder views and the coherent integration of multiple practice elements consistent with TR principles that contribute to improved outcomes and positive school stakeholder experience within a specialist context.

### 4.2. Implications for Practice

A major systemic implication of this study is the positioning of student wellbeing *ahead* of academic progress and, likewise, supporting educator wellbeing in practical, ongoing ways. The neuroscience of learning [6,8,10,11,20,28,37,65] has clearly established that higher-order cognitions, such as problem solving and predicting the outcomes of a particular action, require a student's brain and limbic system to be in a learning-ready state, that is, *within* their individual "window of tolerance" for stress [11]. Fundamentally, as our sense of wellbeing (safety, trust, relational connection, and readiness) *declines*, fear and stress responses *increase*, taking the brain "off-line" for learning tasks as it attunes to a real or perceived threat in the environment. Students such as those with generalised anxiety need safe and trustworthy relationships within the school environment to mitigate their already heightened state of survival and narrower window of tolerance (i.e., vulnerability to being triggered), without which educational efforts are compromised by constantly active fear and stress responses [14]. Table S3 illustrates practice elements used at the school that where specifically valued by all participant groups.

The University of Chicago Urban Education Initiative evidences a stronger link between school climate and student social–emotional wellbeing and academic achievement, respectively, than that between academic achievement and student demographics such as ethnicity or economic advantage [66,67]. As Noddings [68] emphasises, the establishing of a climate of care and empathy is not in addition to teaching and learning, "it is underneath all we do as teachers. When that climate is established and maintained, everything else goes better". Aligned to this is supporting the educator characteristics of being empathic and establishing strong relationships. Active educator attunement was here linked to a shift in student self-belief and motivated more positive social engagement, social–emotional development, wellbeing, and learning gains. Embedding wellbeing-focused practices in school systems requires responding to the evergreen challenge of balancing relentlessly increasing demands on educators to create "space" for deeper student connection and building a culture of collective care. These skills of reflection and inquiry are urgently needed within initial teacher training globally [69,70].

From the perspective of educators, there were a number of enablers of the schools' TR culture, including (a) the integration of TR knowledge into policies and practices [50–52]; (b) the need for synergy among TR knowledge, values, and practices of educators; (c) school leadership that is distributive [23] and modelled TR pedagogy and respect across the school community; (d) reflective practice; and relatedly, (e) an inquiry-based process to develop a shared and communicated vision, building trust, and a collaborative culture [67]. Many of these practices are central to Indigenous concepts of learning and teaching that build belonging, sense of place, and the dignity of community members [71–73]. These are practices that need to be explored with Indigenous partners who may be positioned as leaders in TR educational change throughout education systems. By incorporating inquiry-based, collaborative, and equity-focused processes, members of the school community could become mutual learners and teachers working towards shared goals, with a deep appreciation of the need for multiple knowledges and ways of knowing.

### 4.3. Limitations

Several study limitations need mentioning. The school was in its sixth year of building a TR culture, with staffing levels, continuity, and time to co-develop shared approaches which would arguably contribute to positive outcomes, whereas it may be more challenging

in larger, more diverse settings and schools in the early stages of introducing TR practices. A transference of outcomes from this small specialist school to mainstream schools cannot be assumed, given several important distinctions, such as the high teacher–student ratio and resourcing to serve students with high-levels of social–emotional and learning and support needs. Furthermore, the school was positioned within a Catholic ethos of pastoral care, including the guiding belief in servanthood and the value of community. Since 2020, the ReLATE model has, however, been revised, consolidated, and extended into non-specialist Catholic schools. Initial findings from a 3-year evaluation of 15 schools indicate that the pedagogy and practice of ReLATE can translate to practice within general Catholic schools. Further information regarding this larger study, which concludes in April 2024, can be obtained from The MacKillop Institute [41].

## 5. Conclusions

The use of a relational trauma-responsive and social justice-informed pedagogy within the emergent ReLATE model supported positive outcomes and stakeholder experiences. The school culture was positively experienced by students, guardians, educators, and community agencies. The model placed an emphasis on wholistic wellbeing, collective care, and use of reflective inquiry. Trauma was understood as a collective experience requiring a collective process of healing and coherent equitable systems and practices. Perspectives expressed in this study constitute a valuable contribution to understanding what a TR school culture includes. In addition, the findings inform further research and pedagogical development to advance recovery from trauma and improve student academic and wellbeing outcomes. The TR elements of mutuality and collaboration to proactively address and reduce the perpetuation of trauma in the lives of individuals and groups within a school community could be important aspects for future research.

**Supplementary Materials:** The following supporting information can be downloaded at: https://www.mdpi.com/article/10.3390/traumacare3040028/s1, Table S1: Example of interview and focus group questions; Table S2: Additional quotes from stakeholders of the school; Table S3: ReLATE practices mapped to research themes.

**Author Contributions:** Conceptualization, J.C.A. and H.S.; methodology, J.C.A.; software, J.C.A.; validation, J.C.A., E.G. and H.M.; formal analysis, J.C.A.; investigation, J.C.A. and E.G.; resources, J.R.; data curation, J.C.A.; writing—original draft preparation, J.C.A.; writing—review and editing, J.C.A., H.M., J.D., E.G. and J.R.; visualization, J.C.A. and H.M.; supervision, H.S., H.M. and J.D.; project administration, J.C.A. All authors have read and agreed to the published version of the manuscript.

**Funding:** This research received no external funding.

**Institutional Review Board Statement:** The study was conducted in accordance with the Declaration of Helsinki and was approved by Monash University Human Research Ethics Committee (MUHERC)—17826—on 11 February 2019.

**Informed Consent Statement:** Informed consent (assent was obtained from students following caregiver consent for their involvement) was obtained from all subjects involved in the study.

**Data Availability Statement:** Supplementary Tables S1–S3 provide additional data, including "raw data"—participant quotes to contextualise the findings and conclusions presented in this study. Contact the corresponding author regarding other queries.

**Acknowledgments:** We would like to express our gratitude to the participants for their valuable time and input provided for this research study.

**Conflicts of Interest:** The authors declare no conflict of interest or competing financial interests or personal relationships that could have appeared to influence the work reported in this paper.

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
