# Peer review of "Raising Voice at School: Preliminary Effectiveness and Community Experience of Culture and Practice at an Australian Trauma-Responsive Specialist School"

_traumacare, doi:10.3390/traumacare3040028_

Round 1
Reviewer 1 Report
Comments and Suggestions for Authors
Dear Authors:
The work is characterized by its originality and is well structured. Traumatic events have affected the daily lives of many students, directly impacting indicators correlated to their mental health; consequently, it seems perfectly reasonable that it has affected the quality of life of young people.
I consider it relevant to mention that his work is methodologically very well structured, which allows its replication in different educational contexts, well written, with valid methods and relevantly structured results, discussion, and conclusions.
Only one observation that seems important to me, Table 1 could be adjusted to the format of Table 2 to homogenize the presentation.
Author Response
Response:
Thank-you for your kind words which are greatly appreciated. We value the benefits of transparent methodology. We have adjusted the format of Table 1.
Reviewer 2 Report
Comments and Suggestions for Authors
First of all, thank you all for the interesting article. I have some suggestions for your consideration.
- Please provide a reliability score for all six scales, as you did for (ARTIC) Scale
- Please add your analysis for other scales to your analysis 2.4.1. chapter as well. So far, you have provided details for only ARTIC.
- Please add the results of the quantitative analysis to your results section (so far, only the guardian’s SDQ and educator’s results are reported. The rest results section presents findings from school documentation and qualitative data.
- Please elaborate more on why/how “student wellbeing ahead of academic progress” matters and what are the implications of this in academic settings.
Author Response
Response: 1 Thank-you for this comment. In the first submission we referred the reader to Diggins (2020) for more measure details, analysis and discussion of outcomes. We have added the reliability scores for the SDQ based on Australian data and psychometric properties of the Strengths and Difficulties Questionnaire (Hawes and Dadds, 2004; Mellor; 2004), which indicate moderate to strong internal reliability and stability across all SDQ subscales. Coefficient alphas (a) for each of the five SDQ subscales, and the total difficulties and impact scales range from a = .59 (peer problems) to a = .80 (hyperactivity). Adequate validity was evidenced in the relationship of these scales to one another. SDQ scores are predictively valid, evidencing the feasibility of the SDQ as a screening instrument with concurrent validity against diagnostic interviews ranging from 0.12 — 0.57. The coefficient alphas for the SDQ subscales ranged from 0.65 to 0.91 for each of the subscales. McDonald’s omega or Jöreskog rho ranges from 0.67–0.90 for the parent version (Stone et al, 2015)
Response: 2 Thank-you for this comment, the following detail has been added:
‘A large effect size in mean Total Difficulties, a decrease of 29%, indicated a clinically significant improvement in wellbeing and mental health concerns; with the greatest improvement seen for students in their first year at the school. Of the students who were new to the school that year, 37.5% had a diagnosis of ADHD. Hyperactivity in this group decreased at a consistent rate throughout the 12-month period, with the mean hyperactivity score reducing from a classification of ‘clinical’ to ‘slightly raised’. Over 12 months a significant reduction in conduct problems (F = 6.76, p = .04), hyperactivity (F = 11.39, p = .01), and total difficulties (F = 12.07, p < .01) was reported, along with a close to significant reduction in mean scores for emotional symptoms (p = .06); reducing from above a clinical range to below the clinical range postintervention (Diggins, 2020). As the SDQ data has already been published, we refer the reader to Diggins (2020) for a more detailed analysis and discussion. The CEMSIS and EES do not have analytics in the public domain at this time.’
Response: 3. ‘A major systemic implication in this study is the positioning of student wellbeing ahead of academic progress; likewise supporting educator wellbeing in practical ongoing ways. (Dix et al, 2020). The neuroscience of learning [7,8,11,30,38,67] has clearly established that higher order cognitions such as problem-solving and predicting outcomes from a particular action, requires a students’ brain and limbic system to be in a learning-ready state; that is within their individual ‘window of tolerance’ for stress [7]. Fundamentally, as our sense of wellbeing (safety, trust, relational connection, and readiness) declines, fear and stress responses increase, taking the brain ‘off-line’ for learning tasks as it attunes to real or perceived threat in the environment. Students such as those with generalised anxiety need safe and trustworthy relationships within the school environment to mitigate their already heightened state of survival and narrower window of tolerance (i.e.: vulnerability to being triggered); without which educational efforts are compromised by constantly active fear and stress-responses [14]’
The University of Chicago Urban Education Initiative evidence a stronger link between school climate and student social emotional wellbeing to academic achievement, than that between academic achievement and student demographics such as ethnicity or economic advantage [68,69]. As Noddings [70] emphasises, the establishment of a climate of care and empathy is not in addition to teaching and learning, ‘it is underneath all we do as teachers. When that climate is established and maintained, everything else goes better’ (p.777). Aligned to this is supporting educator characteristics of being empathic and establishing strong relationships. Educator active attunement was linked here to shift in student self-belief, motivated more positive social engagement, social-emotional development, wellbeing and learning gains. Embedding wellbeing focused practices in school systems will require responding to an evergreen challenge of balancing relentlessly increasing demands on educators, to create ‘space’ for deeper student connection and building a culture of collective care. These skills of reflection and inquiry are urgently needed within initial teacher training globally [71].
Response: 3. ‘A major systemic implication in this study is the positioning of student wellbeing ahead of academic progress; likewise supporting educator wellbeing in practical ongoing ways. (Dix et al, 2020). The neuroscience of learning [7,8,11,30,38,67] has clearly established that higher order cognitions such as problem-solving and predicting outcomes from a particular action, requires a students’ brain and limbic system to be in a learning-ready state; that is within their individual ‘window of tolerance’ for stress [7]. Fundamentally, as our sense of wellbeing (safety, trust, relational connection, and readiness) declines, fear and stress responses increase, taking the brain ‘off-line’ for learning tasks as it attunes to real or perceived threat in the environment. Students such as those with generalised anxiety need safe and trustworthy relationships within the school environment to mitigate their already heightened state of survival and narrower window of tolerance (i.e.: vulnerability to being triggered); without which educational efforts are compromised by constantly active fear and stress-responses [14]’
The University of Chicago Urban Education Initiative evidence a stronger link between school climate and student social emotional wellbeing to academic achievement, than that between academic achievement and student demographics such as ethnicity or economic advantage [68,69]. As Noddings [70] emphasises, the establishment of a climate of care and empathy is not in addition to teaching and learning, ‘it is underneath all we do as teachers. When that climate is established and maintained, everything else goes better’ (p.777). Aligned to this is supporting educator characteristics of being empathic and establishing strong relationships. Educator active attunement was linked here to shift in student self-belief, motivated more positive social engagement, social-emotional development, wellbeing and learning gains. Embedding wellbeing focused practices in school systems will require responding to an evergreen challenge of balancing relentlessly increasing demands on educators, to create ‘space’ for deeper student connection and building a culture of collective care. These skills of reflection and inquiry are urgently needed within initial teacher training globally [71].
Reviewer 3 Report
Comments and Suggestions for Authors
This article sets out to explore.to explore multiple stakeholders views of 16 core elements of trauma-responsive practice at a specialist school, the impact on student wellbeing and learning outcomes, and insights into implementation challengers and enablers.
It does indeed uniquely incorporate four participant cohorts: parents or caregivers, educators, and community agencies involved with school students and their families, in a mixed methods approach with an emphasis on the voice of participants and lived experience of a trauma-responsive specialist school, an which is to be commended, demonstrating the valuing of the actual experiences of the people involved, which is very much to be welcomed.
In terms of the study, the methods seem appropriate, and findings valuable, including in the areas of how a trauma-responsive school culture can lead to high staff satisfaction, improved student wellbeing and attendance, and progress towards learning goals, centred around a collective experience of the school as a connected community, emphasising relationships, safety, collaboration, mutuality, voice, and empowerment.
There are several areas which I would recommend the authors pay some more attention to, however.
‘Educators are expressing ever-increasing interest in trauma-responsive (TR) school- 52 wide approaches as they seek to improve learning and wellbeing outcomes for students impacted by adversity and the adults that support them’- This does need more evidencing/ referencing, as it is an important point to jump off from as to why the work was undertaken to evaluate such an approach in a specialist school such as this. A paragraph or two on how this type of approach might be being used (or not) in the wider school system will be valuable, and if it is not being used, some of the reasons why. This also relates to a point that I think should be emphasised more in the limitations and discussion sections, about how such an approach may be valuable in small schools such as this, with a very high staff student ratio, and how this might be transferred to other such small specialist schools, such as the one evaluated here, but also how, if at all (or why not) this approach could be used in other schools which are much larger, and/or maybe present other challenges than this particular one does. Whilst there is a small section on this- ‘A transference of outcomes from this small specialist school to mainstream schools cannot be assumed given several important distinctions such as the high teacher-student ratio, and resourcing to serve students with high-levels of social-emotional and learning and support needs. Furthermore, the school was positioned within a Catholic ethos of pastoral cared?.’, this I think can be strengthened.
I would also ask clarification about one of the statements- ‘-Almost three out of four children in Australia (similarly US and UK) have been exposed to potentially traumatic events (Emerging Minds, 2020) including 12.1 per cent of children (in Victoria) who were the subject of one or more substantiations of family harm such as exposure to violence, neglect, or abuse (Child Protection Australia, 2021); I think the definition of potentially traumatic events needs clarification/expanding, and also how this relates to poverty exactly?
In relation to the section- ‘Environments aiming to support wellbeing, healing, and post-traumatic growth and prevent further harm must: a) realize the widespread impact of trauma and potential paths for healing; …. and vi) responsive to cultural, historical and gender issues. Translating TR principals (principles?) ’, whilst there is brief mention made in the tables of the number of 'Indigenous Australian'/ 'Aboriginal' children- and I did wonder if this should/ could be made more consistent throughout?- I think, as these are such major issues for children in relation to trauma, whether there could be a few more sentences setting out these issues in relation to other work in the field, which relate to the researchers’ own findings.
As I was reading it several. Grammatical issues arose, and one more of those I noticed is as follows:
‘The younger the infant or child, the higher the risk and degree of developmental harm from trauma, such as effecting brain architecture and social responsivity’
Comments on the Quality of English LanguageNeeds some reworking- examples in the comments
Author Response
Response: 1. Thank-you for your comments.
In response to your first point, we have added citations and statements regarding systemic trauma, disenfranchised students, and use of TR schools:
‘Children and young people around the world are reporting ever increasing mental health concerns and effects of adversity including increased exposure to domestic violence, community violence, and social inequities such as poverty, food-insecurity, and homelessness [1,2,4,5,10,20]. Systemic inequities arising from bias, colonisation, and knowledge sovereignty perpetuate, and increase the trauma load for students already disenfranchised by ‘othering’, related to characteristics of race, disability, sexuality, historicity [15]. Australian students at highest risk for trauma exposure are Indigenous, in child protection or youth justice services, or are refugees/asylum seekers [4]. Trauma elevates neurobiochemical stress responses throughout the body. These can remain active and toxic throughout adulthood, adversely impacting health, mental health, relationships, employment and contribute to poverty and homelessness [5,6]. The younger the infant or child, the higher the risk and degree of developmental harm from trauma, such as effecting brain architecture and social responsivity [7].’
Lines 143-146 expand on the purpose of the study: ‘This study responded to the urgent need to address the potentially life-long impact of childhood adversity and the global increase in the mental health and wellbeing needs of students and their educators [8,9,11,28] through the implementation of relational, neuro-science informed, and trauma responsive practices.’
Response 2:
When referring to the potential restrictions regarding generalisability we have added:
‘Since 2020 the ReLATE model has however, been revised, consolidated, and implemented in 3 other specialist primary schools and adopted by non-specialist Catholic schools. Initial findings from a 3-year evaluation of 15 Catholic general schools indicate that the pedagogy and practice of ReLATE in specialist campuses can translate to generalist Catholic schools with support from ReLATE trainers and coaches. Further information regarding this larger study which concludes April 2024, can be obtained from the MacKillop Institute [76]’.
Response 3: Thank-you for identifying the need for clarity here. This paragraph now reads: ‘The term ‘potentially traumatic’ acknowledges that adversity is experienced within the context of the individual, such that the presence of adversity does not necessarily predict whether or not an individual experiences a negative impact or the degree of impact. Individual context includes supports, relationships, personal strategies, historicity, and strengths that interact uniquely for individuals. Contextual issues such as poverty or systemic injustice add to the allostatic load (accumulation of stressors) carried by students and communities, that impedes wellbeing.
Response: Thank-you for identifying the need for clarity here. This paragraph now reads: ‘The term ‘potentially traumatic’ acknowledges that adversity is experienced within the context of the individual, such that the presence of adversity does not necessarily predict whether or not an individual experiences a negative impact or the degree of impact. Individual context includes supports, relationships, personal strategies, historicity, and strengths that interact uniquely for individuals. Contextual issues such as poverty or systemic injustice add to the allostatic load (accumulation of stressors) carried by students and communities, that impedes wellbeing.
Response:
Thank-you for your observations, we see the potential for confusion. In this manuscript the terms ‘Aboriginal and Torres-Strait Islander’ is used when referring specifically to the First Nations people of Australia, while ‘Indigenous’ refers more generally to global First Nations peoples. We have added to the sentence at 704: ‘Many of these practices are central to Indigenous concepts of learning and teaching around the world’, to clarify this distinction. Lines 70-74: Trauma-informed initiatives began in adult mental health and health care [12] and have since been adapted and adopted across health, education, and social care services over the past two decades. Recent TR models in education are centering relationality, addressing inequities, and responding to systemic, racial, cultural, and historic trauma [1,5,6,8,10,11].
Response: We appreciate having this bought to our attention, corrections have been made.
Round 2
Reviewer 2 Report
Comments and Suggestions for Authors
Thank you for considering the reviewers's suggestions.
Reviewer 3 Report
Comments and Suggestions for Authors
Thank you for your revisions, and we can now move to final processes for the paper